# Frequency, Informativity and Word Length: Insights from Typologically Diverse Corpora

**DOI:** 10.3390/e24020280

**Published:** 2022-02-16

**Authors:** Natalia Levshina

**Affiliations:** Max Planck Institute for Psycholinguistics, 6525 XD Nijmegen, The Netherlands; natalia.levshina@mpi.nl

**Keywords:** Zipf’s law of abbreviation, frequency, informativity, n-grams, corpora, linguistic typology

## Abstract

Zipf’s law of abbreviation, which posits a negative correlation between word frequency and length, is one of the most famous and robust cross-linguistic generalizations. At the same time, it has been shown that contextual informativity (average surprisal given previous context) is more strongly correlated with word length, although this tendency is not observed consistently, depending on several methodological choices. The present study examines a more diverse sample of languages than the previous studies (Arabic, Finnish, Hungarian, Indonesian, Russian, Spanish and Turkish). I use large web-based corpora from the Leipzig Corpora Collection to estimate word lengths in UTF-8 characters and in phonemes (for some of the languages), as well as word frequency, informativity given previous word and informativity given next word, applying different methods of bigrams processing. The results show different correlations between word length and the corpus-based measure for different languages. I argue that these differences can be explained by the properties of noun phrases in a language, most importantly, by the order of heads and modifiers and their relative morphological complexity, as well as by orthographic conventions.

## 1. Introduction

One of the most famous generalizations in linguistics is Zipf’s law of abbreviation [1,2]. It states that more frequent words tend to be shorter than less frequent words. For example, highly frequent words like *it*, *go* and *nice* are shorter than *entity*, *locomote* and *agreeable*. This law has been tested in corpora of 986 languages from 80 different families, and a negative correlation between frequency and length was observed in all of them, demonstrating that the law of abbreviation is an exceptionless language universal [3]. From the perspective of information theory, the law of abbreviation is regarded as a manifestation of compression [4]. Cognitively, the negative correlation between frequency and length is likely to be a result of a mostly unconscious process of shortening a linguistic form when its meaning is highly accessible [5]. This is a manifestation of communicative efficiency [6,7]. Experimental evidence shows that learners of an artificial language optimize form-meaning mappings, choosing shorter forms for more frequent meanings under pressure for accuracy and pressure for saving time and effort [8].

In this sense, the law of abbreviation fits nicely with other examples of efficient linguistic behaviour, such as the use of zero or short pronominal forms to refer to accessible referents [9], omission of complementizers when the complement clause is highly predictable from the matrix verb [10] and phonological reduction of words and syllables predictable from their neighbours [11,12,13]; see overviews in [14,15,16]. In many cases, accessibility depends on linguistic context (more exactly, co-text). Importantly, if a linguistic unit is frequently predictable and reduced, this information can percolate into its mental representation. As a result, it is reduced in contexts where it is less predictable, as well [17,18]. Conversely, if a word is not predictable on average, it is also not reduced in predictable contexts [19]. A commonly used term for average contextual unpredictability is informativity, coined by Cohen Priva and Jurafsky [20].

Therefore, we can expect that lengths of words are determined by their informativity. This is also what we could expect based on information theory: the length of a signal in an efficient code depends on its predictability from context, not its frequency [7]. This was the main message of [21], a groundbreaking study by S. Piantadosi, H. Tily and E. Gibson, which showed that length of words in eleven Indo-European languages is more strongly correlated with the words’ informativity than with their frequency. Informativity was based on conditional probabilities of words given one, two or three preceding words. However, a more recent study in [22] has demonstrated that the dominance of contextual informativity is no longer observed if other methodological choices are made: in particular, if one encodes strings in UTF-8, rather than in the ASCII standard, and excludes words that do not occur in spelling dictionaries and in the OPUS corpus of film subtitles [23]. Concretely, only 2 of the 11 Google 1T n-gram datasets used in [21], namely, those representing English and French, show higher correlations of informativity and word length after the dictionary and subtitles filters have been applied to the target words. Yet, German, Italian and Russian show higher correlations of informativity with length based on additional Google Books 2012 datasets, which were not used in the original study.

These results make one wonder if the correlations between length and informativity, and between length and frequency, vary randomly across languages and datasets, or if there is a pattern. To put it differently, are there “frequency-sensitive” languages and “informativity-sensitive” languages? There are some reasons why this question is worth asking. First of all, if wordforms in a morphologically rich language carry inflectional morphemes, these morphemes may or may not be predictable from neighbouring wordforms, depending on the distance between the head and the dependent syntactic element. Second, languages with richer morphology will have a higher type–token ratio, because content words will have many different forms. This can make n-grams based on wordforms less helpful for prediction, because of the probabilities will be less reliable due to data sparseness. In addition, if a language has flexible word order, as is frequently the case with morphologically rich languages, we can expect neighbouring words to be less reliable cues for a target word than in a language with rigid word order.

In both studies of word informativity discussed above, the main focus was on Indo-European languages from three major genera: Germanic, Romance and Slavic (see [24] for some additional languages). In the present study I investigate whether word length is more strongly correlated with informativity than with frequency in a sample of more typologically and genealogically diverse languages: Arabic, Finnish, Hungarian, Indonesian, Russian, Spanish and Turkish.

In addition to examining the diverse languages, I will also compare informativity given previous word and given next word. In fact, it has been found that backward-looking informativity based on bigrams was more strongly correlated with word length than forward-looking informativity in quite a few languages [24]. This may sound counter-intuitive, but in fact many studies of word reduction in speech production [11,13,18] demonstrate that predictability based on upcoming context can be even more strongly correlated with phonological reduction than predictability based on previous context, which means that language users apparently monitor “backward” predictability, as well.

Finally, a methodological goal of this study is to extend the cleaning method in [22] to context n-grams and check if the bigrams selected for analysis lead to different correlations of the resulting informativity measure with word length.

In what follows, Section 2 presents the corpus data and the methods of processing the target and context words and computing the corpus-based measures. Section 3 describes the correlations between these measures and word length. In Section 4, an interpretation of the results is offered. Finally, Section 5 summarizes the findings and provides an outlook.

## 2. Materials and Methods

For this investigation, I used a large web-based corpora of seven typologically and genealogically diverse languages with different writing systems (Arabic, Cyrillic and Latin script with diverse diacritic signs). The data were taken from the Leipzig Corpora Collection [4]. Each corpus contained 10 million sentences collected from diverse online sources and randomly shuffled by the corpus creators. In accordance with [22], UTF-8 encoding was used to map binary codes into human-readable characters. UTF-8 dynamically encodes each character with the minimally necessary number of bits, choosing either 8, 16 or 32 bits depending on the character. It can therefore code more different orthographic symbols than the original or extended versions of ASCII with 7 or 8 available bits, respectively (see more on this in [25]). As a result, UTF-8 codes correctly diacritic marks and non-Latin characters. In particular, it treats strings such as Spanish *si* ‘if’ and *sí* ‘yes’ as different words, which is crucial for correct estimation of their frequency and informativity [22].

The sentences were first tokenized, such that all wordforms were separated by a space from punctuation marks, except for email addresses, fractions, time expressions and other special cases. Hyphens and apostrophes were allowed to be inside a token (e.g., *ice-cream* and *don’t*). Next, I created a list of all unigrams for every corpus. Uppercase characters were converted to lowercase. After that, the unigrams were cleaned; following the filtering methods in [22], I retained only those strings that occurred in the corresponding OpenSubtitles corpus (for this purpose, I used pre-compiled word frequency lists from the repository https://github.com/hermitdave/FrequencyWords/tree/master/content/2018 (last access 29 November 2021)) [26] and in the Hunspell dictionary (with the exception of Finnish, for which the Microsoft spellchecker was used). The removed strings represented words written in a different language (most importantly, English), misspellings, proper names, acronyms and punctuation marks. The total number of tokens before and after the cleaning procedure is provided in Table 1.

From those cleaned lists, I selected 10,000 most frequent words, which served as target words in the subsequent analyses. This was done in order to ensure that there were enough contexts for computation of the informativity measures, following [21]. The length of the target words was computed as the number of UTF-8 characters, with the exception of punctuation within a word (that is, hyphens and apostrophes were not counted). The frequencies were not transformed into self-information (also known as surprisal based on non-conditional probability of a word), because it did not make a difference for computation of Spearman’s rank correlations whether original or transformed frequencies were used. In the subsequent analyses (see Section 3), the frequencies are negatively (log-)transformed, however, but this was done solely for the purposes of visualization and comparability with the informativity measures.

To compute informativity, two sets of bigrams were extracted from each corpus. The methods were as follows:

Method 1: All bigrams were collected, including punctuation and numbers. Spelling errors and other noise were not excluded. This method is similar to the one applied in [22].

Method 2: Only bigrams containing the cleaned unigrams with a frequency of at least 10 were taken into account. Numbers and punctuation marks were included. The low-frequency unigrams were excluded in order to avoid unrealistically high probabilities due to data sparseness, since no smoothing was performed, similar to [21,22].

The two different methods were employed in order to test how robust the results are with regard to different options for data processing. As shown in [22], different methods can lead to dramatically different results, but in that study, only the target words were cleaned. In the present study, I investigate what happens if we apply a similar filter also to context words.

Using the different sets of bigrams, I computed two sets of informativity measures based on one preceding word (or another string), and two sets of informativity measures based on one following word (or another string). The general formula of informativity for a target word *w*, which occurs in total *N* times in a set of diverse contexts *C* is given in (1), following [21].
(1)$$\bar{I}=-\frac{1}{N}\;\sum_{i=1}^{N}\log_{2}P\;(W=w\;|\;C=c_{i})$$

Note that using bigrams, and not higher order n-grams, creates favourable conditions for the prominent role of informativity. Only in this case, [21] found no exceptions in the dominance of informativity in their study, while increasing the context leads to more exceptions and smaller differences between frequency and informativity. There is also a practical reason for testing bigrams only: the modest size of the datasets, in comparison with the previous studies, does not encourage us to estimate informativity based on trigrams and especially not for 4-g.

Word length in characters may not adequately represent the articulation effort in spoken communication, which is crucial for our estimation of efficiency. This is why I performed additional analyses based on word lengths in phonemes. The phonemic representations were obtained with the help of the XPF translation rules [27], which can be conveniently used online via a graphical user interface (https://cohenpr-xpf.github.io/XPF/Convert-to-IPA.html (last access 11 February 2022)). The rules were available for four languages from the sample: Hungarian, Indonesian, Spanish and Turkish. The XPF translation rules were used in [27]. Affricates and long vowels and consonants were counted as one phoneme. The results are presented in next section.

## 3. Results of the Correlational Analyses

First, Spearman’s rank correlations between word length in UTF-8 characters and the three predictability measures (negative frequency, informativity given previous word and informativity given next word) were computed. The correlation coefficients together with their 95% confidence intervals based on all bigrams (Method 1) and on the cleaned bigrams (Method 2) are displayed in Figure 1. Since the method does not affect the correlations between frequency and length, the bars representing negative frequency are identical.

Figure 1 shows that using the cleaned bigrams decreases the correlations between informativity and length in nearly all cases. Negative frequency is consistently positively correlated with length, as predicted by Zipf. Only in Indonesian does informativity given previous word consistently have a stronger correlation than frequency. In Finnish and Hungarian, frequency “beats” both informativity measures. In Russian, the small advantage of previous-word informativity disappears in the cleaned data, such that the difference between correlation coefficients based on informativity and frequency is no longer statistically significant. Notably, the correlations between informativity based on previous word and length are negative for both methods. In Arabic, Spanish and Turkish, informativity given next word is more strongly correlated with length than frequency, while the correlations between informativity given previous word and length are close to zero or even negative (in Arabic).

Statistical significance of these differences was assessed by using two-tailed tests for dependent groups with the help of the method implemented in [28]. Table 2 displays the results of testing the differences between informativity and frequency correlations with length. The tests support the previous observations.

The relationships between different corpus-based measures and length are presented in greater detail in Figure 2. The informativity measures are based on the cleaned bigrams. The points represent trimmed means (based on 500 bootstrapped samples for each length value). The error bars represent the standard deviations from the means. Frequency is negatively log-transformed, in order to facilitate the comparison with the two informativity measures. The curves exhibit some non-monotonicity in the very short and very long words, which is likely to be due to uncertainty, as one can see from the large standard deviations. This uncertainty can be explained by lower type and token frequencies of very long words, and high ambiguity potential of very short words (e.g., *a* is the indefinite article, but also a letter used as a symbol and in lists). Interestingly, Indonesian has substantial non-monotonicity for lengths less than five characters. An explanation of this fact is that Indonesian native lexicon contains mostly words that are disyllabic or longer [29], so native words are quite long. Short words are often borrowings, e.g., Dutch loan words *bir* ‘beer’, *dok* ‘dock’, *dus* ‘douche’ and *bel* ‘bell’. This remarkable phonotactic peculiarity makes very short words less frequent and predictable than one would expect, and slightly longer words more frequent and predictable.

Other non-monotonic patters are observed for Arabic (informativity given previous word) and Russian (informativity given next word), which explains why these languages have negative correlations. In both cases, the informativity scores first rapidly increase, and then, after the length of four or five characters, they gradually decrease. Possible causes of this pattern will be discussed in Section 4.2.

Since the orthographic characters do not map one-to-one onto the actual phonemes, I performed additional correlational analyses using word lengths in phonemes in four languages. The method of obtaining word lengths was described in Section 2. The results of correlation analyses are shown in Table 3, together with the correlations based on the orthographic characters. The informativity measures used for the correlations are based on cleaned bigrams, but analyses based on all bigrams show a very similar picture. The differences between the correlation coefficients are very small. This means that the previous conclusions are not affected.

## 4. Interpretation of the Correlations

### 4.1. Different Methods of Processing Bigrams

The correlation analyses reveal that the cleaned bigrams tend to yield lower correlations between informativity and length than all bigrams. The only exception is Finnish, where the correlation of informativity given next word becomes actually slightly stronger. In Russian, the correlation between length and informativity given previous words becomes similar to the correlation between length and frequency, when the cleaned version of bigrams is used. In Indonesian, frequency becomes clearly more important than informativity given next word. These findings extend the results in [22]: put together, the analyses show that cleaning both target words and context words leads to a lower status of informativity.

But how can we explain these small but persistent differences between the methods? I have looked at the Russian data more closely, focusing on informativity based on previous word, and computed the absolute difference between the words’ ranks depending on length and their ranks depending on informativity for the data with all bigrams. This difference was called a ‘rank gap’. Next, the same procedure was performed for the cleaned data. Finally, I computed the difference between the two rank gaps for every target word. A positive difference means that informativity based on the cleaned bigrams better fits the length-based rank of a word. A negative difference means that the data with all bigrams produce a better fit. I have examined the target words with particularly large differences, and their bigrams.

A major factor that boosts informativity seems to be the high proportion of hapax legomena and low-frequency words, which occur exclusively or predominantly with a target word in the data with all bigrams. For example, 40% of all bigrams occurring with the wordform *xrame* ‘temple’ in the Locative case predict it with the probability of 1. These are low-frequency words, which are adjectives formed from the names of the saints, e.g., *svjato-olginskom* ‘St. Olga’, or other rare proper names specifying the location or religious attributes of the temple. Other examples of bigrams with proper names as context words are *rusgrain holding* ‘the holding Rusgrain’, *krasnojarskomu kraju* ‘to the Krasnoyarsk region’, *kirovskij rajon* ‘the Kirov district’ and *ejfelevoj bašni* ‘of/from the Eiffel tower’. Some context words that have been filtered out are in fact existing technical terms, which are extremely unlikely to occur in film subtitles, e.g., *mineralovatnyje plity* ‘mineral-wool slabs’, *mašinostroitelnyje zavody* ‘machine-building factories’, *usb porta* ‘of the/a USB-port’. In nominal phrases like these, the target word is short and represents a generic term, while the adjective or noun makes it more specific. When these context words are removed, the short head noun becomes less predictable, which decreases the correlation between informativity and length.

Spelling errors are also responsible for higher predictability of some short target words. In some cases, parts of one word are written as two words, probably as a result of word division due to hyphenation. As a result, the second part is short and highly predictable, e.g., *toč no* instead of *točno* ‘exactly’, *graž dan* instead of *graždan* ‘of (the) citizens’. The second part is ambiguous with real words (e.g., *no* ‘but’ and *dan* ‘given’), which explains why it was included in the list of target words. All this means that the method of data processing can lead to important differences in the results, even when the target words are the same. We also see that the cleaning method can filter out words that are real noise and words that are perfectly rare and domain-specific, but valid.

### 4.2. Cross-Linguistic Variation in the Effects of Informativity

An unexpected finding is that Arabic and Russian have negative correlations of length with some of the informativity measures. In particular, Arabic has a strong negative correlation between length and informativeness given previous word. A closer look at the data suggests that this unexpected result has to do the structure of definite nominal phrases. The lexical modifier, which follows the head noun, is highly predictable given the head noun, but it is long because it carries the definite article. Moreover, relational adjectives derived from nouns carry derivational morphology suffix *-iyy(a)*, as in the following examples of bigrams:
(1)Arabic

a.  الاتحاد الأوروبي

  al-ittiḥād al-ʾūrubbiyy
  DEF-Union DEF-European
  ‘the European Union’ 
b.  الموقع الإلكتروني

  al-mawqial-iliktrōniyy
  DEF-site DEF-electronic
  ‘the website’


This creates a mismatch between low informativity and substantial length. Yet, we cannot exclude that the writing conventions in Arabic, where vowels are not represented graphically, can play a role in explaining these findings. Testing this requires a very large transcribed corpus.

In Russian, I observed a negative effect of predictability given next word. This can be because short words are often nouns in the genitive case, which occur at the end of the noun phrase. They often have zero marking in plural. Examples are *nauk-Ø* ‘science.PL.GEN’, as in *akademija nauk* ‘the Academy of Sciences’, *ptic-Ø* ‘bird.PL.GEN’, as in *penije ptic* ‘birds’ singing’, *bljud-Ø* ‘dish.PL.GEN’ as in *bolšoj vybor bljud* ‘a wide selection of dishes’. These forms are short, but are difficult to predict from the following token, which can explain the high informativity scores for and four- and five-character words in Figure 2.

At the same time, there are many pre-nominal adjectives which form set expressions and are therefore highly predictable from the noun. They often have denominal derivational morphology and agree with the noun in case, number and gender. This makes them morphologically complex and quite long, which contributes to the negative correlation between length and informativity given the following noun, and the non-monotonic pattern in Figure 2. Below are examples of bigrams with such adjectives:
(2)Russian

a.  *electron-n-oj**počt-y*
  electron-ADJ-GEN.SG.Fpost-GEN.SG 
  ‘of electronic mail’ 

b.  *plastik-ov-yx**okon-Ø*
  plastic-ADJ-GEN.PLwindow-GEN.PL
  ‘of plastic windows’

c.  *otečestv-enn-oj**vojn-e*
  fatherland-ADJ-DAT/LOC.SG.Fwar-DAT/LOC.SG
  ‘to/about/in the Patriotic War’

d.  *buxgalter-sk-ogo**učet-a*
  accountant-ADJ-GEN.SG.Mbookkeeping-GEN.SG
  ‘of accounting’


The structure of the noun phrase and morphology of adjectives can also explain why Spanish and Turkish have near-zero correlations between length and informativity given previous word. In Spanish, the order is usually noun + adjective, where the adjective agrees with the noun in gender and number. Examples are given in (3). Since adjectives are often predictable from the noun, we would expect them to be short. This is not the case, however, due to the presence of derivational and inflectional morphemes, which weakens the effect of informativity given previous word.
(3)Spanish

a.  *sensor**infrarroj-o*
  sensor.SG.M infrared-SG.M
  ‘infrared sensor’
b.  *línea*
*puntead-a*
  line.SG.F dotted-SG.F
  ‘dotted line’
c.  *caldera-s*
*industrial-es*
  boiler-PL industrial-PL
  ‘industrial boilers’
d.  *aire**acondicionad-o*
  air.SG.M conditioned-SG.M
  ‘air conditioning’

In Turkish, the noun phrase has the opposite properties, but, somewhat paradoxically, this leads to the same outcome. First of all, the order in the noun phrase is Modifier (adjective or noun) + Noun. Second, the noun carries the case and number marking, while the modifier is bare. Consider some examples in (4). In this situation, the noun can be highly predictable given the modifier, but it can carry many inflectional suffixes.
(4)Turkish

a.  *Birleşmiş*
*Millet-ler-’den*
  unitednation-PL-ABL
  ‘from the United Nations’
b.  *Kültür**bakanlığ-a*
  cultureministry-DAT.SG
  ‘to the Ministry of Culture’
c.  *komiser*
*yardımcı-sı*
  comissioner helper- 3.SG.POSS
  ‘deputy commissioner’

Thus, in Spanish and Turkish, the less complex first element (a head noun in Spanish and adjective or modifying noun in Turkish) is followed by a more complex second element (an adjective in Spanish or head noun in Turkish). This can also explain why informativity given next word has a positive correlation with length. In Russian, the adjective in the Adjective + Noun phrase is more complex than the noun. Plural genitives in the Noun + Genitive phrases are also often less complex than the head noun. This is a possible reason for why informativity given previous word is positively correlated with length in Russian.

These impressions are supported by a more systematic quantitative analysis. For reasons of time and computation effort, I sampled randomly 10,000 sentences from every corpus. The sampled sentences were then parsed with the help of the *udpipe* package in R [30]. I extracted all common nouns, when they were modified by other common nouns or adjectives in immediate proximity to the head. The Universal POS tags used for this purpose were ‘amod’ (adjectival modifier), ‘nmod’ (nominal modifier of a noun), ‘appos’ (apposition) and diverse multiword expressions represented by the dependencies ‘compound’, ‘fixed’ and ‘flat’ (See https://universaldependencies.org/u/dep/index.html (last access 1 December 2021)). I extracted the lengths of nominal heads and modifiers in UTF-8 characters.

The estimated mean lengths and standard deviations are shown in Table 4. Only the data for the predominant order are given. The data suggest that if the first element (head or modifier, depending on the language) is longer than the second one, as in Indonesian and Russian, then informativity given previous word has a higher correlation with length than informativity given next word. If the second element is longer, as in Arabic, Finnish, Hungarian, Spanish and Turkish, then informativity given next word has a higher correlation with length than informativity given previous word. Interestingly, the longer heads or modifiers also tend to have greater standard deviations than the shorter ones (except for Spanish).

In general, the structure and morphology of the noun phrase seem to determine the strength and direction of correlations of length with different measures. This is not surprising, since nouns are by far the most frequent types and tokens in the corpora. The proportion of tokens (wordforms) with the Universal POS tag “NOUN” relative to all tokens, excluding punctuation, ranges from 24.3% in Spanish to 48% in Arabic. The proportion of types with the tag “NOUN” is from 33.4% in Spanish to 55.8% in Turkish. From this follows that the correlations in the wordform samples depend very strongly on how predictable and predictive, and how long or short nouns are.

Of course, this hypothesis rests on the assumption that all modifiers are predictable and/or predictive with regard to their neighbouring heads. This assumption is not ungrounded. First of all, the principle of information locality states that words that strongly predict each other should be close to each other in linear order [31]. Second, although it is unfortunately impossible to determine the structure of the corpora in terms of genres and text types, an informal inspection suggests that the sentences come mostly from informative texts (e.g., online news, product advertisements and reviews, educational texts and IT manuals), in which one finds formal and technical phrases with tightly associated heads and lexical modifiers.

### 4.3. Frequency or Informativity?

The last and perhaps the most exciting puzzle is why frequency is more strongly correlated with length than any of the informativity measures in Finnish and Hungarian. These languages seem to have a plenty of predictable but long words, which makes the informativity measures less strongly correlated with length.

Let us examine informativity given next word, which shows higher correlations with length than informativity given previous word in both languages. Applying the method used in Section 4.1 (comparison of rank gaps), it is possible to identify target words in which frequency outperforms informativity in predicting the length rank of a word. In Finnish, one finds many adjectives, which are long, but occur in relatively fixed expressions, which makes them highly predictable from the head noun, e.g., *seksuaalisen häirinnän* ‘of sexual harassment’, *ammatillinen koulutus* ‘vocational education’, *ortodoksisen seurakunnan* ‘of the Orthodox Church’ and *kolmannelle osapuolelle* ‘to a third party’. In Hungarian, there are many denominal adjectives formed with the *-i* suffix, which are quite complex morphologically, e.g., *csatlakozási tárgyalások* ‘accession negotiations’, *tartózkodási engedélyt* ‘residence permit’, *külkereskedelmi mérleg* ‘foreign trade balance’ and *természettudományi múzeum* ‘natural science museum’. These expressions seem to belong to official and formal discourse.

As for informativity given previous word, one class of words that can contribute to the relative weakness of this correlation is postpositions, e.g., Hungarian *keresztül* ‘through, across’, *érdekében* ‘for the benefit of’, *kapcsolatban* ‘in connection with’, *kapcsolatos* ‘in relation to’ and *köszönhetoen* ‘thanks to’. They are highly predictable because they are preceded by a certain case form of the noun. They are often long because they have complex morphological structure, e.g., *érdekében* consists of *érdek* ‘interest’, possessive suffix *-e* and inessive suffix -*ben* ‘in’. This is strikingly different from complex prepositions in English, which are written as separate words.

Note that Indonesian, in which informativity plays a clearly dominant role, has particularly many dependencies marked as compounds. Examples are *proses akreditasi* ‘accreditation process’, *pintu tol* ‘toll gate(s)’ and *hari kerja* ‘working day(s)’. There is no morphological marking. The nouns are just stung together, similar to English, but in reverse order. They are also written separately. Since they are predictable, informativity plays a prominent role in Indonesian, as it does in English, according to [22]. Therefore, derivational and inflectional morphology and spelling conventions are responsible for the relationships between frequency and informativity.

These are not the only factors, of course, which can explain the observed cross-linguistic differences in the role of frequency and informativity. For example, native and frequent words in Indonesian tend to be polysyllabic, which can be a factor that inhibits the correlation between frequency and length in Indonesian. We should also not discard the potential role of high type–token ratio in highly inflectional languages such as Finnish and Hungarian. As argued in Section 1, high type–token ratio is expected to result in less reliable estimates of wordform-based predictability. To support this claim, I created 1000 samples of type–token ratio values for a random sample of 1 million tokens from every corpus (only the cleaned unigrams were considered). The mean type–token ratios were computed then. As expected, Finnish and Hungarian have the highest type–token ratios (0.123 and 0.116, respectively), whereas Indonesian has the lowest value (0.021). The other languages have values in between: Spanish (0.043), Arabic (0.092), Turkish (0.097) and Russian (0.099).

Finally, it was hypothesized that the flexibility of word order can also influence the results. I extracted average word order entropies in simple clauses using data from a previous study [32]. The entropies were based on the probabilities of different orders of syntactically related words (a head and its dependent, or two co-dependents). The probabilities were extracted from the Universal Dependencies corpora. A high entropy value means more flexible word order. According to the data, Finnish has the highest average entropy (0.603), followed by Russian (0.578), Hungarian (0.501) and Spanish (0.499). Arabic (0.417), Indonesian (0.395) and Turkish (0.341) have lower average entropies. This mixed picture suggests that word order, as represented by average entropy, cannot differentiate between the frequency-sensitive and informativity-sensitive languages very well. However, in order to make a final decision we need a more nuanced analysis, which looks only at the adjacent dependencies and heads which constitute n-grams. Therefore, the question about the role of word order flexibility remains open.

## 5. Conclusions

This study examined correlations between word length and frequency, informativity given previous word and informativity given next word. Seven languages with different typological properties, genealogical origin and writing systems were examined: Arabic, Finnish, Hungarian, Indonesian, Russian, Spanish and Turkish. The data were obtained from large web corpora from the Leipzig corpora collection. The target words were the most frequent 10,000 unigrams that were found in spelling dictionaries and film subtitles. Two types of bigrams were used for the computation of informativity: (1) all possible sequences of two tokens in a corpus; and (2) only bigrams in which both tokens were found in a spelling dictionary and film subtitles and occurred at least 10 times in the corpus, plus numbers and punctuation marks.

First of all, the method of processing bigrams for computing informativity plays some role: in nearly all cases, we see that the correlation between informativity and length becomes slightly weaker if the informativity measure is based on the cleaned bigrams. In particular, in Russian, this correlation is no longer significantly different from the correlation between frequency and length. This is in line with the findings in [22], who showed a similar decrease in the prominence of informativity after the target unigrams have been cleaned.

Second, the data show that informativity is not the measure most strongly correlated with word length in all languages. In Finnish and Hungarian, simple frequency outperforms each informativity measure, regardless of the method of bigram processing. This supports the results obtained in [22]. The dominance of informativity is thus not universal. Clearly, we need to take into account informativity based on trigrams and 4-g, but at least we see that the results are not uniform. In contrast, the correlations between frequency and length are robust cross-linguistically, supporting [1,2,3].

Third, the effects of informativity given previous word and next word vary substantially across the languages. While in Indonesian and Russian, informativity given previous word has a higher positive correlation with length than informativity given next word, the other languages in our sample show that informativity given next word is more strongly correlated with length than informativity given previous word. This does not contradict efficiency theory: we can also save effort by shortening linguistic units that are predictable given upcoming cues, as known from numerous studies of predictability effects in language production [11,13,18]. However, we also find negative and near-zero correlations between some informativity measures and length, against all expectations.

Explaining these results is not easy, but a few hypotheses have been formulated based on samples of sentences annotated with the Universal Dependencies. The results are likely to be due to the cross-linguistic differences in the structure and morphology of the noun phrase, because nouns represent a very significant proportion of the target and context words. The status of informativity given previous and next word depends on the relative length of nouns and their lexical modifiers in the noun phrase, which are strongly associated and therefore mutually predictive. Namely, if the shorter element in a noun phrase is followed by the longer one (shorter + longer), informativity given next word will be more strongly correlated with length because the first element is both short and highly predictable. If the order in the noun phrase is reverse (longer + shorter), then informativity given previous word is more strongly correlated with length because the second element is predictable and short.

As far as the dominance of frequency in Finnish and Hungarian is concerned, it can be explained by substantial length of predictable elements (most importantly, adjectives) in formal and technical noun phrases, due to rich derivational and inflectional morphology. Indonesian, the least morphologically complex language in the sample, displays the most obvious dominance of informativity. We can thus expect that the role of frequency and informativity is dependent on nominal and adjectival morphology, as well as on orthographic conventions; while Indonesian has many nominal compounds that are written as separate words, they are often written as one word in Finnish and Hungarian, such that the mutual predictability of their elements can no longer be captured by an n-gram approach. Since English is close to Indonesian with regard to compounding and the poverty of morphological marking, this can explain why informativity is systematically prominent in English, regardless of the methods and datasets [22]. Therefore, analytic languages are more likely to be informativity-sensitive, whereas synthetic languages are more likely to be frequency-sensitive.

Importantly, these cross-linguistic differences are likely to be observed in informative text types with many noun phrases representing special terms and set expressions, e.g., in online news, user’s manuals or educational texts. Unfortunately, it is very difficult to compare the genre structure of the corpora, since the sentences have been reshuffled and the sources are not available. It is possible that word lengths in more informative corpora will be more informativity-sensitive, while word lengths in less informative texts will be more frequency-sensitive. I leave this hypothesis to future research.

Moreover, we cannot exclude that high type–token frequency in highly synthetic languages makes the informativity measures less reliable. All these claims require testing in the future.

In view of these findings, what can we say about the role of communicative efficiency in determining word length? The picture is quite complex. First of all, wordforms, which are the unit of analysis in the present study, carry both lexical and grammatical meanings. The latter often participate in paradigmatic contrasts, where one meaning (and corresponding form) is considered unmarked, and the contrasting meaning (form) is considered marked. Examples are singular vs. plural, present vs. future, positive vs. comparative degree of comparison [33]. These asymmetries have been explained in terms of frequency and predictability: for example, the singular (uniplex) meaning is more frequent and therefore more predictable, which is why it is usually not formally marked, or marked by a shorter morpheme than the plural (multiplex) meaning [34,35]. How lexical predictability interacts with grammatical predictability is not fully understood yet.

Moreover, we cannot assume that language is processed in the brain as n-grams, determined by orthography. Grammatical change is driven by the processes of chunking and automatization of articulation routines, when a sequence of words is frequently processed together to express one meaning [36]. This is why estimation of predictability from n-grams may not the best approximation of real accessibility of forms and meanings for a language user. Although the use of corpus frequencies is also not unproblematic, due to the same orthographic conventions, and also because our expectations of a word depend on the subject domain and genre [16], corpus frequencies seem to be more robust. At the moment, the positive correlations of informativity with length cannot be regarded as *direct* evidence for communicative efficiency, and the negative correlations cannot serve as direct evidence against it. How exactly local predictability can percolate into language structure and, in particular, determine word length, remains thus an open question.

## Figures and Tables

**Figure 1 entropy-24-00280-f001:**
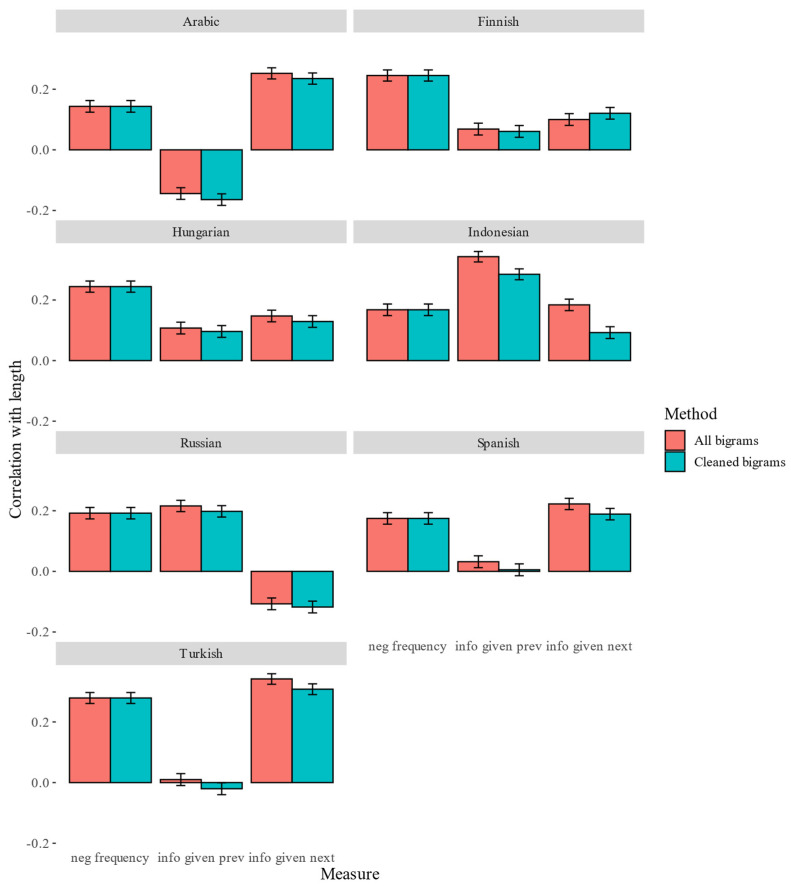
Spearman’s correlations between length and corpus-based measures in seven languages. Pink—based on all bigrams; turquoise—based on cleaned bigrams.

**Figure 2 entropy-24-00280-f002:**
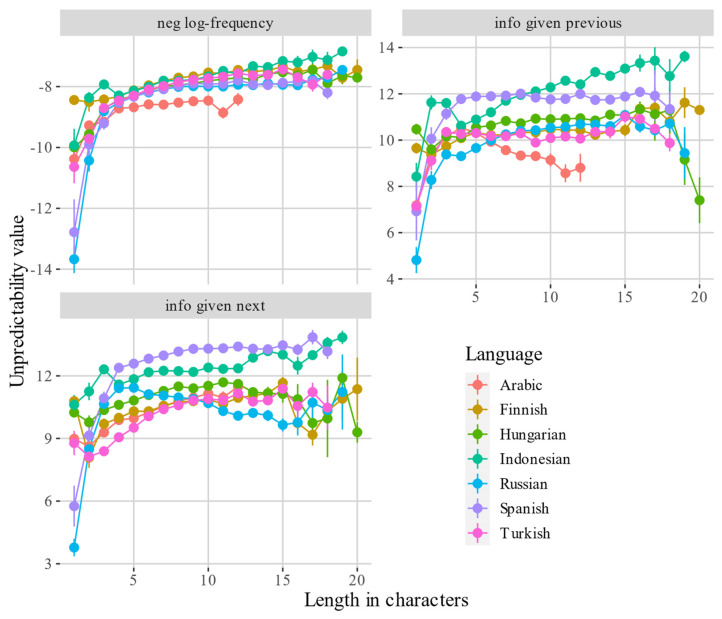
Relationships between word lengths in characters and three corpus-based measures: negative log-frequency (**top left**), informativity given previous word (**top right**) and informativity given next word (**bottom left**). The points are the trimmed means based on 500 bootstrapped samples. The error bars represent one standard deviation from the mean. The informativity measures are based on cleaned bigrams.

**Table 1 entropy-24-00280-t001:** Genealogical information (according to www.wals.info, accessed on 1 December 2021) and number of tokens in the web corpora.

Language	Genus	Total Number of Tokens (with Punctuation Marks)	Number of Tokens in Dictionaries and Subtitles
Arabic	Semitic	232,612,052	196,210,964
Finnish	Finnic	130,150,353	93,230,461
Hungarian	Ugric	192,639,921	139,922,959
Indonesian	Malayo-Sumbawan	183,318,963	131,109,405
Russian	Slavic	165,441,482	127,190,505
Spanish	Romance	206,340,525	168,109,866
Turkish	Turkic	157,739,375	108,278,142

**Table 2 entropy-24-00280-t002:** P-values of testing the differences between correlation coefficients (informativity vs. frequency). In parentheses, the winning measure with the highest positive correlation with length is displayed.

Language	Bigrams Processing Method	Frequency vs. Informativity Given Previous Word	Frequency vs. Informativity Given Next Word
Arabic	all bigrams	*p* < 0.0001 (frequency wins)	*p* < 0.0001 (informativity wins)
	cleaned bigrams	*p* < 0.0001 (frequency wins)	*p* < 0.0001 (informativity wins)
Finnish	all bigrams	*p* < 0.0001 (frequency wins)	*p* < 0.0001 (frequency wins)
	cleaned bigrams	*p* < 0.0001 (frequency wins)	*p* < 0.0001 (frequency wins)
Hungarian	all bigrams	*p* < 0.0001 (frequency wins)	*p* < 0.0001 (frequency wins)
	cleaned bigrams	*p* < 0.0001 (frequency wins)	*p* < 0.0001 (frequency wins)
Indonesian	all bigrams	*p* < 0.0001 (informativity wins)	*p* = 0.05 (not significant)
	cleaned bigrams	*p* < 0.0001 (informativity wins)	*p* < 0.0001 (frequency wins)
Russian	all bigrams	*p* = 0.017 (informativity wins)	*p* < 0.0001 (frequency wins)
	cleaned bigrams	*p* = 0.54 (not significant)	*p* < 0.0001 (frequency wins)
Spanish	all bigrams	*p* < 0.0001 (frequency wins)	*p* < 0.0001 (informativity wins)
	cleaned bigrams	*p* < 0.0001 (frequency wins)	*p* = 0.017 (informativity wins)
Turkish	all bigrams	*p* < 0.0001 (frequency wins)	*p* < 0.0001 (informativity wins)
	cleaned bigrams	*p* < 0.0001 (frequency wins)	*p* = 0.002 (informativity wins)

**Table 3 entropy-24-00280-t003:** Correlations between predictability measures and word length in phonemes and in characters: Spearman’s correlation coefficients and 95% confidence intervals (in parentheses). The informativity measures are based on cleaned bigrams.

Language	Predictability Measure	Correlation with Length in Phonemes	Correlation with Length in Characters
Hungarian	Neg. frequency	0.25 (0.23, 0.27)	0.24 (0.23, 0.26)
	Info given prev	0.10 (0.08, 0.11)	0.10 (0.08, 0.12)
	Info given next	0.14 (0.12, 0.16)	0.13 (0.11, 0.15)
Indonesian	Neg. frequency	0.16 (0.14, 0.18)	0.17 (0.15, 0.18)
	Info given prev	0.28 (0.26, 0.30)	0.28 (0.27, 0.30)
	Info given next	0.09 (0.07, 0.11)	0.09 (0.08, 0.11)
Spanish	Neg. frequency	0.17 (0.15, 0.19)	0.17 (0.16, 0.19)
	Info given prev	−0.01 (−0.03, 0.01)	0.01 (-0.01, 0.02)
	Info given next	0.18 (0.16, 0.20)	0.19 (0.17, 0.21)
Turkish	Neg. frequency	0.28 (0.26, 0.30)	0.28 (0.26, 0.30)
	Info given prev	−0.02 (−0.03, 0.00)	−0.02 (−0.04, 0.00)
	Info given next	0.31 (0.29, 0.34)	0.31 (0.29, 0.33)

**Table 4 entropy-24-00280-t004:** Mean lengths and standard deviations of common nouns and their lexical modifiers. Numbers in bold show the longer NP elements. The numbers are valid only for the predominant order.

Language	Predominant Order	Number of Modified Heads	Mean Length of Heads	Mean Length of Modifiers
Arabic	Head + Modifier	39,200	5.07 (SD 1.58)	**5.96** (SD 1.71)
Finnish	Modifier + Head	11,603	**9.53** (SD 4.17)	9.08 (SD 4.07)
Hungarian	Modifier + Head	18,358	**8.73** (SD 3.5)	7.68 (SD 3.31)
Indonesian	Head + Modifier	17,257	**6.70** (SD 2.43)	6.62 (SD 2.39)
Russian	Modifier + Head	13,057	7.27 (SD 2.71)	**9.37** (SD 3.28)
Spanish	Head + Modifier	6759	7.69 (SD 2.58)	**8.42** (SD 2.58)
Turkish	Modifier + Head	19,246	**7.04** (SD 3.56)	5.95 (SD 3.1)

## Data Availability

The data presented in this study are available in the Appendix A.

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
