# Peer review of "Frequency, Informativity and Word Length: Insights from Typologically Diverse Corpora"

_entropy, 2022, doi:10.3390/e24020280_

Round 1

Reviewer 1 Report

This study looks at the relationship between word length and predictability in a diverse sample of languages, containing Arabic, Finnish, Hungarian, Indonesian, Russian, Spanish, and Turkish. The author shows that word length vs. several surprisal measures is fairly different across languages. Unlike much of the prior literature, the author digs into the details of the syntax and word order of some of these languages in order to present some plausible explanations for the cross-linguistic patterns. In general, this is an important result and dataset and I recommend publishing it with some minor modifications. 

One strength of this paper, especially compared to the prior literature, is that the author looks at a cleaned corpus and is attentive to the fact that the reliability of the measures depends on frequency. However, I would recommend looking in more detail at the way in which reliability changes with frequency (and predictability) by using a split-half correlation. This reliability should be taken into account when computing the correlations. (This is also related to the hapax legomena phenomena the author discusses)

Additional comments:

- Figure 1 might be better with some baseline comparisons -- what size correlations should we expect based on other studies?

- Figure 2 might be easier to read if ti was broken down by language. I'm not sure what the error bars are, but SDs might be more useful than standard errors? This figure also has a somewhat weird aspect ratio (Figure 1 does too I think) but in Figure 2 it  might be fixed by faceting by language.

- Table 3 might be nicer as a figure? I'm not sure.

Author Response

Dear Reviewer, I am very grateful for your constructive and useful comments!

I have improved the aspect in Figure 1 and Figure 2 and changed the error bars in Figure 2, which now represent standard deviations. Many thanks for this suggestion. I have also added an explanation of the error bars, which was missing.

I fully agree that reliability is crucial. It definitely deserves a separate study, probably, with more advanced statistical methods than Spearman’s correlations. I used this simple method for better comparability with the previous studies, but we need to go beyond that in future studies.

Unfortunately, I did not understand the suggestion on split-half correlations, probably because I have never used them before. I've found some papers on them after reading your review, but still have difficulties imagining what the design would look like. 

I've tried plotting Figure 2 by languages, but it was suboptimal because of the different y-values for the three measures, which made it impossible to show clearly the non-monotonic patterns I wanted to demonstrate.  Still, thank you for this suggestion. I hope with the improved aspect it is easier to see what I wanted to show.

I have considered adding previous correlations as reference points, but unfortunately the previous papers use different methods and different sets of languages and corpora, so the results are not directly comparable, in my view.

Reviewer 2 Report

Review of "Frequency, informativity  and word length" by Natalia Levshina

Review by Michael Cysouw <[email protected]>. Please note that I do *not* want to remain anonymous!

== General impression ==

This paper is a nice combination of some statistical results with some in-depth analysis of individual languages. Such a combination is rare. The paper is well-written, except for the introduction, which is confusing.

My main reservation with the paper is that there is no clear distinction between methods and dicussion. The main result (in my opinion) is presented in section 4, and even in the conclusion that are new figures computed (type/token ratios). There should be a stronger distinction between methods and discussion. Also, less emphasis should be placed on the cleaning of data.

The results in Figure 1 suggest that the difference between the cleaned and non-cleaned correlation are not very substantial (contrary to 173-174). This part of the paper to me seems not to be the most intriguing result. Maybe put less emphasis on this data cleaning in the introduction and the method. In contrast: also include the method of the analyses in section 4 at the start of the paper.

Section 4 is introduced as an "interpretation of the results" (99) but actually there is new data presented here (Table 3). This table is actually one of the more intriguing results of the paper, and it should be included in Section 3. It should also be announced in the introduction. The match between Table 2 and Table 3 looks like a significant result that should be highlighted (see suggested in the running commentary below).

== Running commentary ==

The abstract and the introduction are stylistically much less clear in comparison to the rest of the paper. I suggest to revise these parts substantially. I will add some suggestions here:

Abstract

- 1-8 "at the same time" This phrase is used regularly in this paper. Try to use something else, like "however", "in contrast"
- 1-10 "can be more strongly" => is more strongly correlated with word length than with frequency
- 1-11 remove "which"
- 1-12 remove "in"
- 1-13 "Turkish)," close sentence here and start anew "This research shows/argues..."
- 1-13 "intriguing" This word is empty, say explicit what is found, or simply leave out this word
- 1-13 "differences" what differences? maybe use something like "different correlations for different languages"
- 1-17 "consistent cross-linguistic differences" maybe something like "different correlations for different languages" (yes, it is repetitive here, consider revising)
- 1-19 add "by" to read "by the order of heads"

Introduction

The first two paragraphs try to relate a whole range of different concepts, viz. frequency, compression, efficiency, accessibility, predictability, and informativity. The relation between those concepts is very interesting, but their relation remains rather cryptic in this introduction. Try to make it clearer, maybe also remove a few of those concepts (to remain short), specifically compression and accessibility seem superfluous in the present context.

- 1-32 "problem" why is compression a problem for coding theory?
- 1-34 remove "the" before "speakers"
- 1-34 "if" => "when"
- 1-37 remove "combined"
- 1-41 remove "in cases"
- 1-44 "verbal" => "linguistic"
- 2-49, 2-53 remove the names, or simply use surnames
- 2-57 "at the same time" => however
- 2-61 to 65 rephrase these complete sentences. Something like: Concretely, only two of the 11 Google 1T n-gram datasets used, viz. those representing Eng and Fre, show higher correlations between informativity and word length than between frequency and word length (no comma here!) after  the filters have been applied. (the following sentence does not make sense to me: rephrase!)
- 2-72 what is the "length" of syntactic dependency? And why should this matter? please explain
- 2-75ff The topic of flexible word order is not discussed in this paper. Consider leaving it out of the introduction
- 2-81 "if" => whether
- 2-82 reorder comparison "word length is more strongly correlated with informativity than with frequency"
- 2-88ff Even after reading these sentences I will found the phenomenon counter intuitive. The text here does not make it less counter intuitive to this reader. Maybe explain a bit more?
- 2-94 "we" ? There appears to be only a single author

Materials/Methods

- 3-107 remove "The" before UTF-8
- 3-110 what kind of "tokenization" was performed?
- 3-Table 1: maybe add a column with percentage reduction aber cleanup? Arabic and Spanish seem to have less of a reduction after cleanup. Is there a reason for this?
- 3-159 add "especially not for 4-grams"

Results

- 3-163 name the three other measures here
- 4-173 "decreases to correlations" can you be more specific about how strong/significant this decrease is? Just from visual inspection the difference does not seem to be very large in most cases. To be clear: the difference between the cleaned and the non-cleaned correlations does not seem to be strong.
- 4-Table 2 Maybe highly the structure of the result, e.g. highlight those cells where informativity wins. Consider ordering Arabic with Spanish and Turkish because they show the same direction of the correlations. The languages seem to fall into three groups (Fin/Hun, Ind/Rus and Ara/Spa/Tur). Maybe add stronger lines between those groups.
- 5-Figure 2 This figure does not help me understand what is going on in the data. Maybe more explanation? Maybe remove the figure?

Discussion

- 8-269 "possible reason" This turns out to be a hypothesis that is later tested in this paper, so it is not "possible" but "actual" :-) 
- 8-287 "writing conventions in Arabic" Consider Maltese, which uses a latinate orthography  
- 9-366 here starts a new subsection. Or consider adding all of the following to the method and results section.
- 10-376 "Only data for the predominant order are given" This is unclear: does this mean that the numbers in Table 3 are only based on one order for each language?
- 10-381ff Remove this last sentence of the paragraph. It does not add anything. If anything, it seems like the SD are very high. It might be more interesting to test whether there is any difference between the mean lengths. For example, the difference for Indonesian is surely not significant.
- 10-384 "determine" If you really argue for determination, please try some kind of predictive modeling!
- 10-Table 3 Please add some statistical comparison between the mean lengths (not just boldface for the highest number) Also: Highlight clearly in the table that Indonesian and Russian are different from the others!!! Because of the way the table is structured this is not obvious (I had to go through the numbers one by one to see this highly important result!)
- Add a table or figure to show the correlation between Table 3 and Table 2. This is a really interesting result. Ind/Rus have larger first element, Fin/Hun have longest average elements. This is exactly the same grouping as in Table 2.

Conclusion

- 12-455ff This effect is not very strong and not clearly established in the paper.
- 12-490 Fin/Hun have the longest words overall of all the languages studies in this paper.
- 13-514 reverse "we should also not"
- 13-519 The type-token ratios should not be added in the conclusion. Add these data to an early section.

Author Response

Dear Michael,

many thanks for your thoughtful and detailed comments!  First of all, Sections 4 and 5 have been restructured, such that the additional data analysis required for interpreting the correlations is presented in Section 4, and Section 5 contains only the conclusions. 

Following one of your comments, a discussion of word order variability is added at the end of Section 4, with some entropy scores from the Universal Dependencies corpora.

I have also added a reference to Moran and Cysouw (2018) and a very brief explanation of the differences between UTF-8 and ASCII (Section 2), as requested by the editor. I hope my explanation is correct.

I refer to Figure 2 more often in the text, so I hope that it is more justified now (this was one of your concerns). 

The tokenization method is now briefly explained in Section 2, as well. 

I’m very grateful for the suggestion to test Maltese, which has similar grammar to Arabic, but a different writing system. This would be extremely interesting, but requires a large corpus and plenty of time. Computing the informativity scores is very time-consuming, even with our MPI computer cluster.

I have considered adding previous correlations as reference points, but unfortunately the previous papers use different methods and different sets of languages and corpora, so the results are not directly comparable, in my view.

Unfortunately, I have not saved all lengths of heads and modifiers in the annotated data. As a result, I cannot immediately perform a significance test of the length differences in Table 4, as you request. If the editor finds it necessary, I will be happy to rerun the analyses and test these differences, but I would need extra time for that.  

I really appreciate your suggestions about the interpretability of the numbers and figures in the part about the typological differences in the NP, but I would rather keep the original alphabetic order of the languages in the tables and in the pictures. I hope the interpretation given in the text is clear enough.

I have implemented numerous stylistic suggestions from your review, as you can see with TrackChanges (I do hope they will be visible after I submit the new version). I’m very grateful for all these comments. They have helped me to notice and correct many imperfections.

Best wishes,

Natalia

Reviewer 3 Report

The manuscript is a contribution to Zipf's law of abbreviation, which
is a tendency of words to be shorter when they are more frequent. As
shown by S. Piantadosi, H. Tily and E. Gibson in 2011, a stronger
correlation than between the word length and the word frequency occurs
between the word length and the word informativity. The informativity
is the average minus logarithm of the conditional probability of the
word (given either preceding or succeeding words). Thus, there is a
plausible tentative connection between word lengths and the lengths of
optimal codewords in information theory. There may be also a tendency
to make the stream of information as constant as possible given other
constraints (such as inevitable peaks of unpredictability at the
boundaries between words).

The present manuscript investigates the correlation between word
lengths and informativity estimates for seven typologically diverse
languages (Arabic, Finnish, Hungarian, Indonesian, Russian, Spanish
and Turkish). What is an interesting contribution of the reviewed
manuscript, sometimes the backward informativity is stronger
correlated with length reduction than the forward
informativity.---Roughly speaking, language users seem to shorten a
given word in anticipation of words that are yet to be uttered rather
the words that had been already produced. The exact experimental
results vary across different languages. Among three predictors of the
word length, being frequency, forward informativity, and backward
informativity, there are languages that any of them is the best.

As the author rightly observes, the exact conclusions can be
interfered by imperfectness of the estimates of
informativity. Conditional probabilities in our minds and hence
informativities are not directly observable. At best, they can be
approximated by simpler or more complex probability models. The author
chose very simple estimates of conditional probability---given by
unsmoothed word bigram frequencies.

It would be interesting to check in the future research whether the
conclusions for some languages change when more advanced statistical
language models are used. In particular, in the case of agglutinative
languages such as Hungarian and Finnish, where the type/token ratio is
very high, the bigram estimates can be prone to overfitting. Then it
is not surprising that it is exactly those languages for which the
bigram informativity is the worse predictor.

In the future research in this vein, it would be advisable to better
control for statistical significance of conclusions. Frankly speaking,
I am not sure whether the standard tests for Spearman's correlation
can be meaningful when one of the random variables (a simple
informativity estimate) is an imperfect version of the "ground truth"
(a better informativity estimate). It is a nontrivial task to devise a
better statistical methodology. One should not only compute
informativities but also take into account the precision up to which
they are estimated. Maybe some Monte Carlo simulations and a
collaboration with statisticians could help?

Author Response

Dear Reviewer,

many thanks for sharing your thoughts on this exciting topic! You highlight the importance of estimating reliability of informativity measures. I fully agree that reliability is crucial. It definitely deserves a separate study, probably, with more advanced statistical methods than Spearman’s correlations. I used this simple method for better comparability with the previous studies, but we need to go beyond that in future studies.